# Adversarial Robustness of Pruned Neural Networks

**Luyu Wang, Gavin Weiguang Ding, Ruitong Huang, Yanshuai Cao, Yik Chau Lui**
Borealis AI
Toronto, ON M5G 1L7, Canada
`{luyu.wang,gavin.ding,ruitong.huang}@borealisai.com`
`{yanshuai.cao,yikchau.y.lui}@borealisai.com`

## Abstract

Deep neural network pruning forms a compressed network by discarding "unimportant" weights or filters. Standard evaluation metrics have shown their remarkable speedup and prediction accuracy in test time, but their adversarial robustness remains unexplored even though it is an important security feature in deployment. We study the robustness of pruned neural networks under adversarial attacks. We discover that although pruned models maintain the original accuracy, they are more vulnerable to such attacks. We further show that adversarial training improves the robustness of pruned networks. However, it is observed there exist trade-offs among compression rate, accuracy and robustness in adversarially trained pruned neural networks. Our analysis suggests that we should pay additional attention to robustness in neural network pruning rather than just maintaining the classification accuracy.

## 1 Introduction

Recently, the high demand for mobile-friendly deep learning models necessitates the compression of such models (Cheng et al., 2017). Weight pruning is a simple yet effective compression method (Han et al., 2015b; Guo et al., 2016; Li et al., 2016; See et al., 2016; Molchanov et al., 2016). It usually proceeds in three stages: first a full network is trained as usual; then "unimportant" weights are pruned according to some criteria, resulting in a sparse model with lower accuracy; finally, the sparse model is fine-tuned to recover the original accuracy. Moreover, filter pruning for CNNs is proposed to obtain models that are more compatible with the hardware (Li et al., 2016; Molchanov et al., 2016). Pruning has the advantage over other compression methods that it produces models with high compression rate and high accuracy. It is widely applied to computer vision and natural language processing systems (Han et al., 2015a; See et al., 2016).

Orthogonal to the neural network pruning literature, it is well-known that deep models have the vulnerability that they can be fooled to make wrong predictions on inputs that are carefully perturbed (Szegedy et al., 2013). One explanation is that deep neural networks, despite very complex architectures, contain mostly linear operations; this allows one to easily craft adversarial examples from the first order approximation of the loss function (Goodfellow et al., 2014). Many stronger adversarial attacks are also proposed (Moosavi-Dezfooli et al., 2016; Papernot et al., 2016; Carlini & Wagner, 2017), as well as the internal representation of a neural net (Sabour et al., 2015). These attacks typically require the gradient information from the target model to be attacked; however, the so-called black-box attack (Papernot et al., 2017) only requires knowing the predictions from the target model. Adversarial training has been shown effective defending against adversarial attacks (Huang et al., 2015; Kurakin et al., 2016; Madry et al., 2017). A more robust model can be obtained when adversarial examples are included in the training procedure.

In this paper, we try to answer the following questions: *what are the effects of model pruning on the robustness of the model against adversarial attacks? Can we build an adversarially robust model that have both high compression rate and accuracy?* We believe these are important because of the popularity of compressed models - their security risk needs to be understood. To the best of our knowledge, this paper is the first to investigate the interplay between neural network pruning

and adversarial attacks/training. We discover that pruned networks with high compression rates are more vulnerable to adversarial attacks. We also study the effectiveness of adversarial training on pruned models, and find there exists trade-offs among the classification accuracy, compression rate, and adversarial robustness.

## 2 VULNERABILITY OF PRUNED NEURAL NETWORKS

We use a three-layer CNN [1] trained on MNIST (LeCun et al., 1998) to investigate pruning's influence on adversarial robustness. We also experimented on CIFAR-10 (Krizhevsky & Hinton, 2009) and observed the similar trends (see Appendix A for more information). The original model is first trained with 10 epochs, which is pruned in one shot and fine-tuned for another 10 epochs using RMSprop with a learning rate of 0.001. We focus on the magnitude-based pruning approach, including both weight pruning (Han et al., 2015b) and filter pruning (Li et al., 2016). We study magnitude-based pruning due to its simplicity and effectiveness, and leave evaluation of more complex pruning methods for future works. We adopt the "class-blinded" pruning method introduced in See et al. (2016) for weight pruning, and for filter pruing we use the $l^1$ norm of a filter divided by the number of parameters in the filter to indicate its importance (Li et al., 2016). All weights or filters are ranked across different layers and the small ones are neglected. We consider two types of white-box attacks: fast gradient sign method (FGSM) (Goodfellow et al., 2014) and projected gradient descent (PGD) (Madry et al., 2017), as well as black-box attacks described by Papernot et al. (2016).

Table 1: Accuracy of pruned networks by weight or filter pruning on clean or perturbed test data.

| Parameters pruned | Natural images | FGSM | | | PGD | Papernot's Black-box |
|---|---|---|---|---|---|---|
| | | $\epsilon = 0.1$ | $\epsilon = 0.2$ | $\epsilon = 0.3$ | | |
| 0% | 99.0% | 70.3% | 30.4% | 13.6% | 0.3% | 59.6% |
| weight - 98% | 99.0% | 47.1% | 11.3% | 4.0% | 0.0% | 59.2% |
| filter - 75% | 98.8% | 52.8% | 16.5% | 8.5% | 0.0% | 52.6% |

As suggested in the previous works, we select the smallest model without accuracy loss as the final compressed network. It is seen from Table 1 that, even though performing equally well on the natural images, they are more vulnerable to FGSM and Papernot's black-box attacks than the original network. Moreover, we also craft the attacks by PGD method with a step size of 0.01 for 40 iterations in a $l^\infty$ ball of $\epsilon = 0.3$. None of the models can withstand this stronger type of adversarial attack. It is discussed by Kurakin et al. (2016) and Madry et al. (2017) that adversarial robustness is proportional to the capacity of the model. We suspect that because of pruning, the network's capacity is reduced and thus becomes less robust.

## 3 ADVERSARIAL TRAINING FOR PRUNED NETWORKS AND ROLE OF MODEL CAPACITY

We perform adversarial training along with the network pruning procedure. Perturbations are generated with FGSM (Kurakin et al., 2016) and PGD method (Madry et al., 2017), and both clean and perturbed data are fed into the model in each epoch to ensure high accuracy on the natural images. The $\epsilon$ of FGSM training is sampled randomly from a truncated Gaussian distribution in $[0, \epsilon_{max}]$ with the standard deviation of $\sigma = \epsilon_{max}/2$ (Kurakin et al., 2016). We set $\epsilon_{max} = 0.3$ [2]. PGD training contains examples generated with 40 steps of size 0.01 in a $l^\infty$ ball with $\epsilon = 0.3$.

Table 2 shows that adversarial training is a sound defense strategy for pruned models. It is observed that highly pruned networks is able to become considerably robust even though it is much less parameterized, whereas models with comparable size has previously been shown not trainable adversarially from scratch (Madry et al., 2017). Compared to filter pruning, weight pruning allows more compression because it has greater flexibility on choosing which part of the network to be pruned. Our results also confirm the observations by Madry et al. (2017) that FGSM training is

---

[1]The CNN contains three convolution layers with 32, 64, and 64 filters, followed by a linear layer of 10 units before softmax. The kernel size/stride/padding in each convolution layer is 8/2/3, 6/2/0, or 5/1/0, respectively.

[2]The MNIST data is normalized to the range $[0, 1]$

Table 2: Accuracy of pruned networks by weight or filter pruning on clean or perturbed test data.

| Parameters pruned | Natural images | FGSM | | | PGD | Papernot's black-box | Trade-off |
|---|---|---|---|---|---|---|---|
| | | $\epsilon = 0.1$ | $\epsilon = 0.2$ | $\epsilon = 0.3$ | | | |
| FGSM Training | | | | | | | |
| 0% | 99.2% | 97.9% | 94.0% | 84.7% | 0.5% | 89.2% | - |
| weight - 96% | 99.0% | 94.8% | 83.5% | 59.0% | 2.2% | 79.6% | high compression |
| weight - 80% | 99.2% | 98.2% | 94.7% | 85.9% | 0.2% | 89.6% | high robustness |
| filter - 70% | 98.9% | 94.1% | 82.3% | 60.1% | 1.7% | 82.5% | high compression |
| filter - 60% | 99.0% | 97.8% | 93.6% | 83.0% | 0.4% | 85.7% | high robustness |
| PGD Training | | | | | | | |
| 0% | 99.0% | 97.3% | 95.6% | 93.5% | 92.5% | 96.8% | - |
| weight - 94% | 98.8% | 95.6% | 94.2% | 91.9% | 90.6% | 95.6% | high compression |
| weight - 85% | 99.0% | 96.9% | 95.3% | 93.3% | 92.0% | 96.0% | high robustness |
| filter - 65% | 98.9% | 89.8% | 86.9% | 82.3% | 75.4% | 87.5% | high compression |
| filter - 40% | 99.0% | 94.9% | 93.1% | 90.8% | 87.3% | 94.1% | high robustness |

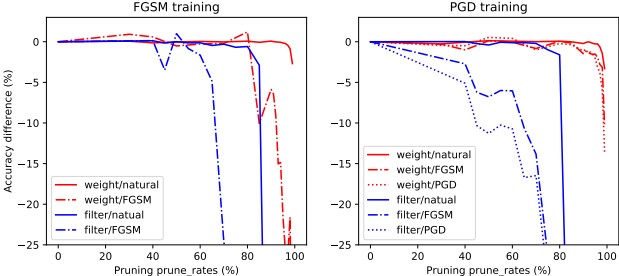

Figure 1: Accuracy differences between the pruned and original model, on both natural and perturbed images. FGSM attacks shown here are with $\epsilon = 0.3$.

suboptimal, and PGD trains more robust models. It is evident in Table 2 that FGSM-training cannot guard against PGD attacks. However, PGD slows down training by a factor equals to the number of iterations used.

It is noticeable in Figure 2 that both the classification accuracy and robustness of the model drop as the compression rate increases. However, robustness drops earlier, which means less compression rate is allowed if we want to maintain the original robustness other than just the original classification accuracy. We suspect the reason is that adversarial training requires additional capacity not just to classify natural data but also separate the $l^\infty$-norm $\epsilon$-ball around each data point. As the network gets pruned more and more, it becomes less capable of modeling such complex decision boundaries. Therefore, the model cannot be highly robust and highly pruned at the same time with FGSM nor PGD training. As shown in Table 2, one can select a highly compressed model but less robust, and vice versa.

## 4 CONCLUSION

We discovered the empirical vulnerability of pruned networks when facing adversarial attacks. Previously, different pruning methods aim at finding the smallest model that maintains the original prediction accuracy. However, we find these severely pruned networks are much easier to be crashed by different types of adversarial attacks. Adversarial training defense strategy is investigated. While pruned models become more robust when adversarially trained, there exists trade-offs among model accuracy, pruning rate, and robustness. High accuracy, high compression rate, and high robustness cannot be achieved at the same time in our experiments on neural network pruning. We suggest that, when security is at stake, model compression should be performed carefully with robustness as a consideration. Our results suggest that to keep a certain model accuracy in adversarial training, one needs to trade-off robustness against compression rate.

ACKNOWLEDGMENTS

The authors would like to thank the discussions with Angus Galloway from University of Guelph.

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

## A ADDITIONAL EXPERIMENTS ON CIFAR-10

We also conduct our experiments using a wide residual network (WRN) (Zagoruyko & Komodakis, 2016) on CIFAR-10 dataset (Krizhevsky & Hinton, 2009) and have observed similar trends. The WRN has 28 layers and a widen factor of 4. We perform both FGSM and PGD training and feed both clean and perturbed data into the model in each epoch. The $\epsilon$ of FGSM training is sampled randomly from a truncated Gaussian distribution in $[0, \epsilon_{max}]$ with the standard deviation of $\sigma = \epsilon_{max}/2$, where $\epsilon_{max} = 8$. The PGD training contains examples generated with a step size of 2 within a $l^\infty$ ball with $\epsilon = 8$ for 7 iterations. Note that on CIFAR-10, adversarial training will reduce the prediction accuracy on the natural images, especially the PGD training.

It is interesting that in the PGD training case when the network is mildly pruned (less than 50% of the parameters), the network is slightly more accurate and more robust than the original one. Meanwhile, when 80% to 94% of weights are pruned, the PGD trained network exhibits higher robustness than the original; however, its classification accuracy drops at the same time. We suspect it is because for such a less parameterized network, more when capacity is allocated on including the perturbed data, there is less for classifying natural images. We leave the analysis on the capacity allocation for future works.

Table 3: Accuracy of pruned WRN-28-4 by weight or filter pruning on clean or perturbed test data.

| Parameters pruned | Natural images | FGSM | | | PGD |
|---|---|---|---|---|---|
| | | $\epsilon = 0.1$ | $\epsilon = 0.2$ | $\epsilon = 0.3$ | |
| Standard Training | | | | | |
| 0% | 95.2% | 66.1% | 53.0% | 42.4% | 0.5% |
| weight - 94% | 94.7% | 54.9% | 33.1% | 24.0% | 0.0% |
| filter - 60% | 94.5% | 49.9% | 30.0% | 21.7% | 0.0% |
| FGSM Training | | | | | |
| 0% | 91.2% | 86.7% | 76.2% | 55.4% | 41.5% |
| weight - 94% | 90.5% | 85.0% | 73.0% | 44.9% | 31.6% |
| filter - 60% | 91.4% | 85.7% | 72.4% | 42.2% | 27.5% |
| PGD Training | | | | | |
| 0% | 87.3% | 83.3% | 74.3% | 55.1% | 47.1% |
| weight - 70% | 87.5% | 83.0% | 74.4% | 55.2% | 47.3% |
| filter - 50% | 87.3% | 83.7% | 75.7% | 55.1% | 47.8% |

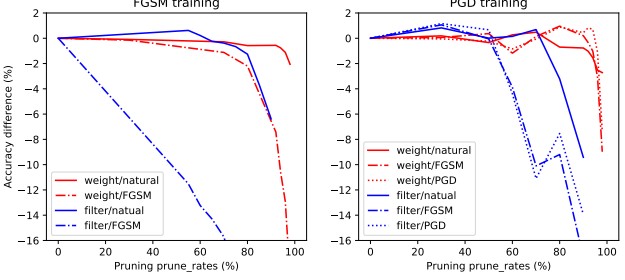

Figure 2: Accuracy differences between the pruned and original WRN-28-4, on both natural and perturbed images. FGSM attacks shown here are with $\epsilon = 8$.

