# OpenReview forum: "Adversarial Robustness of Pruned Neural Networks"
_ICLR.cc/2018/Workshop — Reject_

### Official Review · AnonReviewer3 · 2018-03-07
**Tradeoff between model pruning and adversarial robustness**

**Rating:** 5
**Confidence:** 3

**Review:**

In this work, the authors propose to investigate adversarial robustness of pruned models. Here are some comments:

1 The problem definition seems to be interesting. However, it is not quite novel, since it is a kind of problem with common sense, i.e., when the model is pruned, the capacity is reduced and consequently the robustness is decreased.

2 In Table1, pruning seems to be more sensitive to FGSM. For PGD and Papernot's Black box, the robustness is almost equally bad, w.r.t., models before and after pruning. Does pruning tend to be more sensitive to one particular type of adversarial attacks?

---

> ### Author Response · Authors · 2018-03-22
> **Response to AnonReviewer3**
>
> Thank you for your valuable comments.

---

### Official Review · AnonReviewer2 · 2018-03-09
**Investigation of adversarial robustness of pruned networks**

**Rating:** 5
**Confidence:** 4

**Review:**

The paper performs an empirical investigation of robustness of pruned networks to adversarial samples. They show that pruned networks more susceptible to adversarial perturbations and this can be improved by training with adversarial samples.
Some of the aspects are not covered in this paper :
1) The paper does not give any connection to Stochastic Activation Pruning (SAP) for Robust Adversarial Defense to https://openreview.net/forum?id=H1uR4GZRZ, even though it seems the proposed method is already given in the more elaborate study of various pruning schemes in SAP and its impact on robustness.

2) It is not clear if the model was re-trained after pruning as is suggested in the original paper by Han etal. 2015b. It is possible that the weights need to re-adjusted after pruning, and perhaps the robustness level is maintained compared to the original network.

---

> ### Author Response · Authors · 2018-03-21
> **Response to AnonReviewer2**
>
> Thank you for your valuable comments.

---

### Official Review · AnonReviewer1 · 2018-03-09
**Adversarial Robustness of Pruned Neural Networks**

**Rating:** 6
**Confidence:** 3

**Review:**

In this paper, the authors present a preliminary empirical study of the relationship between neural network pruning and robustness against adversarial examples.

The main conclusion of the study, supported by experiments with small architectures in MNIST and CIFAR 10, is that models which have been pruned according to magnitude-based criteria -- either at the individual weight level or at the filter level -- might be considerably more sensitive to adversarial examples than their unpruned counterparts. Most remarkably, this occurs even if the pruned model retains the same accuracy as the unpruned model on the original, unperturbed data. Therefore, their experiments suggest that overparameterization might be useful in achieving resilience against adversarial examples even if it does not yield a noticeable improvement in accuracy. When adversarial training is used, the results remain consistent with this observation: the higher the compression (i.e. the less overparameterized the model is), the more sensitive the model becomes to adversarial examples.

While the initial findings of this study should be corroborated by a more comprehensive set of experiments, including a wider variety of architectures, larger models, more challenging datasets and more complex pruning algorithms, these preliminary findings might be nevertheless worth discussing as part of the ICLR 2018 Workshop.

MINOR POINTS

The manuscript has many typos and would benefit from additional proof-reading. For example, “pruing”, “suspet” in page 2 or “natual” in Figure 1.

---

> ### Author Response · Authors · 2018-03-22
> **Response to AnonReviewer1**
>
> Thank you for your comments and suggestions.

---

### Decision · Program_Chairs · 2018-03-20
**ICLR 2018 Workshop Acceptance Decision**

**Decision:**

Reject

**Comment:**

Based on the reviews, this paper has not been accepted for presentation at the ICLR workshop. However, the conversation and updates can continue to appear here on OpenReview.